# Comparison of the Efficacy and Safety of Rivaroxaban and Enoxaparin as Thromboprophylaxis Agents for Orthopedic Surgery—Systematic Review and Meta-Analysis

**DOI:** 10.3390/jcm11144070

**Published:** 2022-07-14

**Authors:** Ikhwan Rinaldi, Ihya Fakhrurizal Amin, Yuli Maulidiya Shufiyani, Idham Rafly Dewantara, Brenda Cristie Edina, Kevin Winston, Yusuf Aji Samudera Nurrobi

**Affiliations:** 1Division of Hematology and Medical Oncology, Department of Internal Medicine, Faculty of Medicine, Cipto Mangunkusomo National General Hospital, Universitas Indonesia, Jakarta 10430, Indonesia; 2Faculty of Medicine, Universitas Indonesia, Jakarta 10430, Indonesia; ihyafakhrurizal19@gmail.com (I.F.A.); yulimaulidiya.ym@gmail.com (Y.M.S.); idhamrafly02@gmail.com (I.R.D.); brendacristie@hotmail.com (B.C.E.); kevinwinston4@gmail.com (K.W.); 3Bhakti Medicare Hospital, Cicurug 43359, Indonesia; 4Pertamina Hospital, Balikpapan 76111, Indonesia; yusuf.samudera236s@gmail.com; 5Faculty of medicine, Universitas Airlangga, Surabaya 60132, Indonesia

**Keywords:** rivaroxaban, enoxaparin, orthopedic surgery, thromboprophylaxis, venous thromboembolism, major bleeding

## Abstract

Venous thromboembolism (VTE) is a serious complication that can occur during and after postoperative treatment, including in treatment after orthopedic surgery. The current guidelines for VTE prophylaxis in postoperative patients recommend the use of LMWHs, one of which is enoxaparin. Another recommendation for use in pharmacological VTE prophylaxis is rivaroxaban, which has better efficacy than enoxaparin but a higher bleeding risk. The aim of this systematic review is to provide an update on the profile of rivaroxaban for VTE prophylaxis after orthopedic surgery. PubMed, SCOPUS, EMBASE, and EBSCOhost were searched up until May 2022. The outcome sought was efficacy and safety, described by the incidence of VTE and incidence of bleeding, respectively. Five randomized controlled trials (RCT) were finally included. Rivaroxaban was confirmed to have better efficacy by significantly reducing the risk of VTE and all-cause mortality (RR = 0.38; 95% CI = 0.27–0.54) compared to enoxaparin. However, regarding the safety variable, no significant difference was found between the incidence of major bleeding in rivaroxaban and enoxaparin (RR = 0.97; 95% CI = 0.56–1.68). The results of the analysis show that rivaroxaban has better efficacy than enoxaparin but the same safety profile, so when used, the bleeding of patients should still be monitored.

## 1. Introduction

Venous thromboembolism (VTE) is a serious complication that can occur during and after postoperative treatment, including in treatment after orthopedic surgery. The incidence of VTE is quite high in patients after undergoing orthopedic surgery. For instance, a study on 1,012,823 patients conducted by Inger et al. showed that 7203 of these patients experienced VTE in the first 180 postoperative days, with a cumulative incidence of 0.71% (95% CI, 0.70–0.73) recorded compared with a control group incidence of only 0.11% (95% CI, 0.11–0.12) [1]. Other studies also support this finding of increased VTE. For example, an investigation of VTE in 5783 patients by Viswanath et al. showed that there was a 0.77% rate of symptomatic deep vein thrombosis (DVT) after total hip replacement [2]. Nonetheless, VTE is a preventable cause of in-hospital death. VTE prophylaxis can be administered pharmacologically or mechanically depending on the patient’s bleeding risk [3].

One of the pharmacological prophylaxis medications used for VTE is rivaroxaban. Rivaroxaban is an oral factor Xa inhibitor used at a fixed dose; hence, laboratory monitoring is not required [3]. The recommended dose of rivaroxaban is 10 mg for VTE prophylaxis after elective total knee replacement (TKR) and total hip replacement (THR) [4]. In several phase III clinical trials, rivaroxaban was evaluated as being more effective than enoxaparin in preventing VTE after THR or TKR [5,6,7,8].

The current guidelines for VTE prophylaxis in postoperative patients who undergo total hip replacement and total knee replacement recommend the use of low molecular weight heparin (LMWH), one of which is enoxaparin [9]. Based on the same guidelines, the use of rivaroxaban is also recommended, which has the better efficacy but lacks safety due to the risk of bleeding [9]. However, there are several recent Randomized Controlled Trial (RCT) studies with orthopedic surgery patients that might provide an update on the efficacy and safety profile of rivaroxaban as a prophylaxis. A systematic review study regarding the efficacy and safety of rivaroxaban compared to enoxaparin, conducted in 2012, found that rivaroxaban has a better efficacy than enoxaparin, and concluded that the safety profile of rivaroxaban was at least comparable to enoxaparin [10]. Because the last systematic review and meta-analysis that discusses the use of rivaroxaban in orthopedic surgery was carried out in 2012, an update is needed to determine whether there has been a change in the efficacy and safety profile of rivaroxaban for prophylactic orthopedic surgery. Thus, the aim of this systematic review is to provide an update on the efficacy and safety of rivaroxaban for VTE prophylaxis after orthopedic surgery. Because of more RCTs and, therefore, more patients included in this review than the previous review, more homogeneous data will be provided.

## 2. Methods

### 2.1. Protocol and Registration

Our systematic review was registered in PROSPERO (CRD42022322484).

### 2.2. Ethical Approval

This systematic review is based on research documented in several medical databases. This study did not require any ethics approval.

### 2.3. Literature Search and Data Sources

A systematic computerized literature search was performed in March 2022 to find articles reporting on the efficacy and safety of using rivaroxaban for thromboprophylaxis for orthopedic surgery. The search keywords included terms related to PICO. The following databases were searched: PubMed, SCOPUS, EMBASE, and EBSCOhost.

**Patient**: Adults (>18 years old) who had undergone orthopedic surgery (all types of orthopedic surgery were eligible).

**Intervention**: Rivaroxaban.

**Comparator**: Enoxaparin.

**Outcome**: Efficacy (measured by these variables: (1) incidence of VTE and all-cause mortality, (2) incidence of proximal and distal DVT, (3) incidence of nonfatal or fatal PE, and (4) major VTE (defined as proximal DVT and incidence of PE)) and safety (measured by these variables: (1) all bleeding (major and minor hemorrhage) and (2) major bleeding).

### 2.4. Study Selection

Studies were included in the analysis if (1) they were randomized controlled trials (RCT) on adults admitted for orthopedic surgery, and all types of orthopedic surgery were eligible; (2) they had examined the efficacy and safety of rivaroxaban compared to enoxaparin; (3) they were published in English; (4) the outcomes were the incidence of any bleeding in terms of the safety parameters, and VTE incidence for the efficacy parameters.

Studies were excluded if (1) they were prospective and retrospective cohorts, (2) they were case-control studies, (3) they were case reports, (4) they were case series, (5) drugs other than enoxaparin were used as comparators, (6) they did not use rivaroxaban as the test drug, and (7) their assessment was based on pediatric populations.

### 2.5. Data Abstraction and Risk-of-Bias Assessment

This systematic and meta-analysis was performed in accordance with the 2020 Preferred Reporting Items for Systematic Reviews and Meta-Analyses (PRISMA) guidelines. The keywords used to search the databases are summarized in the table below (Table 1). The titles and abstracts were of all studies from the four databases that matched the keywords, were separately screened according to the eligibility criteria, and duplicates were removed.

Seven reviewers conducted the literature search, screening, and eligibility assessments. Titles and abstracts were separately screened according to the aforementioned inclusion criteria. Furthermore, each study was evaluated for eligibility by seven reviewers, who worked independently and were blinded to each other. Disagreements amongst reviewers were settled by discussions. All of the reviewers extracted data from all of the studies, such as (1) author and year of publication; (2) study design; (3) sample size; (4) characteristics of included patients; (5) intervention details such as dosing and treatment duration; and (6) outcomes presented. The outcomes of our study were divided into two categories: efficacy and safety. We focused on four variables in terms of efficacy: (1) the incidence of VTE and all-cause mortality, (2) the incidence of proximal and distal deep vein thrombosis (DVT), (3) the incidence of fatal and nonfatal pulmonary embolism (PE), and (4) major VTE, which consists of proximal DVT and the incidence of PE. Then, in terms of safety, two factors were investigated: (1) all bleeding (major and minor hemorrhage), and (2) major bleeding. Major bleeding is defined as bleeding that is potentially lethal to the patient and results in a reduction of Hemoglobin (Hb) levels below 2 g/dL based on laboratory evidence. The risk of bias used in the included studies was the Cochrane Risk of Bias 2.0 assessment tool. The appraisal was carried out by seven investigators (I.R., I.F., I.R.D., B.C.E., Y.M., K.W., and Y.A.S.N.), and any differences of opinion were resolved together through discussions.

#### Data Synthesis and Analysis

The data were statistically evaluated, and a meta-analysis was carried out using Cochrane Review Manager 5.4 (Copenhagen, Denmark). Because the original statistics in the study had to be transformed into the Review Manager 5.4 format, there may be minor variations between the original numbers in the study and the table findings from the meta-analysis. The studies were divided into categories depending on whether they used rivaroxaban monotherapy or enoxaparin monotherapy. To represent primary outcomes, the risk ratio was estimated using an inverse variance technique using a random-effects model. To analyze heterogeneity, the findings were given with total values, 95% CI, and I^2^. Low, moderate, and high heterogeneity are indicated by I^2^ values of 25%, 50%, and 75%, respectively, with I^2^ values of more than 50% considered to have considerable heterogeneity. A forest plot was used to help interpret the results.

## 3. Results

### 3.1. Study Selection

The initial search for articles in four databases using specific keywords for each database (Table 1) returned 243 articles, 52 of which were duplicates and removed. Abstracts of the remaining 191 studies were subsequently screened. From the abstract screening, 183 studies were excluded because they did not comply with PICO. Specifically, 117 had non-RCT study designs (were cohorts and review articles), the patient criteria were not relevant (nonorthopedic surgery) in one study, 10 did not use enoxaparin as the comparator drug used, 37 were not using rivaroxaban as an intervention drug, and lastly, in 18 studies, the outcome sought did not match the established PICO (efficacy and safety). The PRISMA flowchart can be seen on Figure A1 of the Appendix A.

A total of eight studies were obtained for full text assessment, and three were excluded for the following reasons: (1) In the RECORD-2 study, the durations of the intervention drug and comparator drug were not the same as those of the other studies (rivaroxaban was given for 39 days [6]. (2) In Kim et al., the two age groups (<60 and ≥60 years old) were given different regimens [11]. (3) In the RECORD-4 study, the dose of enoxaparin given differed from that of other studies (30 mg subcutaneous; meanwhile, the other studies used 40 mg subcutaneous) [8]. Hence, there were five studies that were finally included and appraised in our analysis.

### 3.2. Characteristics of Studies

A summary of the included studies is shown in Table 2. All included studies were randomized controlled trials (RCTs). Prospective cohort studies and case-control studies were not included in this systematic review. The five included studies used the same dose of rivaroxaban, which was 10 mg orally, and the same comparator drug, enoxaparin, at a dose of 4000 IU (40 mg or 0.4 mL) subcutaneously. The duration of treatment varied considerably between the studies, ranging from 15 to 35 days. In total, 8883 patients were included in this review, and the study time range was from 2007 to 2020. The included patients who had mainly undergone orthopedic surgery in the form of total knee replacement, hip replacement, and other nonmajor orthopedic surgeries.

### 3.3. Risk of Bias in Studies and Reporting Biases

We used the Cochrane RoB 2.0 (Copenhagen, Denmark) tool to assess the risk of bias for each of the individual studies. A summary of the assessments is presented in Figure 1.

## 4. Outcome

### 4.1. Efficacy

The five included RCT studies provided the primary outcome of VTE incidence and all-cause mortality, except for the study by Tang et al., which only used VTE incidence as the primary outcome [3,5,7,14]. From the analysis, the total number of events from the primary outcome was found to be 108 out of 4272 (1.2%) patients in the group of patients treated with rivaroxaban. Meanwhile, in the control group (enoxaparin), the primary outcome was 265 out of 4267 (6.2%). Our analysis used a random-effects model to combine the five studies, which were found to have low heterogeneity (I^2^ = 31%). Based on the pooled analysis, it was found that the administration of rivaroxaban could have significantly reduced the incidence of VTE and all-cause mortality based on the obtained risk ratio of 0.38 (95% CI = 0.27–0.54, Figure 2). Furthermore, the incidences of proximal and distal DVT had different distributions: the incidence of distal DVT had a homogeneous distribution, with I^2^ = 0%, whereas the proximal DVT was quite heterogeneous, with I^2^ = 66%. Rivaroxaban significantly reduced the incidence of distal (RR 0.53, 95% CI = 0.41–0.68, Figure 3) and proximal (RR 0.2, 95% CI = 0.05–0.73, Figure 4) DVT. Similar to the incidence of DVT, analysis of the PE incidence showed a heterogeneous distribution of data, with I^2^ = 48%. After pooled analysis, there was no significant difference between the incidence of fatal and nonfatal PE in the rivaroxaban and enoxaparin groups (RR = 0.41 with 95% CI = 0.07–2.38, Figure 5). The last efficacy variable, major VTE, which consisted of proximal DVT and the incidence of PE, showed fairly homogeneous data distribution, with I^2^ = 26%. Rivaroxaban administration significantly reduced the incidence of major VTE compared to the enoxaparin group, with an RR of 0.23 (95% CI = 0.12–0.44, Figure 6) obtained.

### 4.2. Safety Outcome

The five included studies provided the two variables sought for this meta-analysis: (1) any clinically relevant bleeding and (2) major bleeding. The distribution of data for both variables appears homogeneous, with I^2^ = 0% for both variables. The incidence of any clinically relevant bleeding was not different between the rivaroxaban and enoxaparin groups, with a total bleeding incidence of 257 out of 5430 patients for the rivaroxaban group and 243 patients out of 5449 for the enoxaparin group. The risk ratio in the pooled analysis for this variable was found to be 1.07 (95% CI = 0.9–1.27, Figure 7). Furthermore, the same results were obtained for the major bleeding variable. There was no significant difference between the rivaroxaban and enoxaparin groups in terms of the incidence of major bleeding. The incidence of major bleeding was 27 out of 5334 patients in the rivaroxaban-treated group and 27 patients out of 5353 for the enoxaparin group. One study, from Xie et al. [12] reported that there was no incidence of major bleeding in the rivaroxaban and enoxaparin groups. The results of pooled analysis found that the RR for the major bleeding variable was 0.97 (95% CI = 0.56–1.68, Figure 8).

## 5. Discussion

Venous thromboembolism is an acute and potentially life-threatening disease with risk for recurrence [15]. VTE can manifest into DVT and PE. DVT and PE are part of the same syndrome. Risk factors for VTE include cancer, surgery (especially major surgery), trauma, age, and blood disease [16]. In major orthopedic surgery (total knee arthroplasty (TKA), hip fracture surgery (FHS), and total hip arthroplasty (THA)), VTE is more prone to occur in the population undergoing major orthopedic surgery compared to other major surgeries because there are several mechanisms that support the occurrence of VTE in that population. Several prothrombotic processes are also commonly observed, such as the activation of coagulation from tissue and bone damage, reduced venous emptying before or after surgery, immobilization, venous damage, and hot temperatures due to cement polymerisation [17].

In a study conducted by Imberti D et al. for 69,770 patients who had undergone major orthopedic surgery (elective hip and knee replacement), at least 2393 patients experienced VTE events during follow-up, with an average HR of around 3.4% obtained [18]. VTE can occur in orthopedic surgery on the lower extremities. For instance, in a study conducted by Gade IL et al. on patients who had had orthopedic surgery on the lower extremities, a significant incidence of VTE of HR 20.5 (95% CI 17.9–23.5) compared with matched controls was found during a 30-day follow-up. In addition to major orthopedic surgery, minor orthopedic surgery can also increase the incidence of VTE. We also demonstrate that the incidence of VTE in patients undergoing minor distal procedures, such as meniscectomy and arthroscopies, had a significant VTE incidence range of HR 2.9 (95% CI, 1.9–4.4) to HR 7.1 (95% CI 6.4–8.0) [1].

Several guidelines regarding VTE prophylaxis for patients undergoing major orthopedic surgery have been published. The guidelines from the American College of Chest Physicians (ACCP) are said to be more detailed than other guidelines and to provide specific guidance regarding VTE prophylaxis in orthopedic patients. The ACCP guidelines were established based on the type of operation [3].

Based on the ACCP recommendations, patients undergoing THR or TKR may receive LMWHs, low-dose unfractionated heparin (UFH), Vitamin K antagonists (VKA), fondaparinux, apixaban, dabigatran, rivaroxaban, or aspirin in VTE prophylaxis. The prophylaxis should be given for at least 10 to 14 days and can be extended to 35 days. The use of LMWHs is recommended over other agents. The limitations of other agents, such as rivaroxaban, include a lack of long-term safety data and the possibility of increased bleeding [9,19].

The National Institute for Health and Care Excellence (NICE) recommends using LMWHs for 10 days, followed by aspirin for 28 days, or LMWHs for 28 days in combination with antiembolic stockings until the patient is discharged after elective THR. For patients with elective TKR, NICE recommends the use of aspirin at a dose of 75 mg or 150 mg for 14 days or LMWHs for 14 days in combination with antiembolic stockings until the patient is discharged. NICE also recommends the use of rivaroxaban, apixaban, or dabigatran for adult patients undergoing elective THR or TKR. Rivaroxaban may be more effective at preventing VTE than enoxaparin, but it was found to be accompanied by a small increased risk of major bleeding [20].

In patients undergoing hip fracture surgery, the ACCP recommends the use of LMWHs in comparison to other agents. Other agents also recommended are low-dose UFH, VKA, fondaparinux, or aspirin. Prophylaxis is given for 10 to 14 days and may be extended to 35 days [9,19]. In patients with knee arthroscopy, the ACCP does not recommend VTE prophylaxis for patients with no previous history of VTE [9]. In knee arthroscopy patients who are at risk for VTE, the prophylaxes that can be given are LMWHs for 7 to 14 days [20,21].

Rivaroxaban, which is included in the novel anticoagulant (NOAC) group, works through the direct inhibition of factor Xa, which is involved in coagulation [22]. Factor Xa is involved in the common pathway, and its inhibition leads to the prevention of clot formation and thrombin generation [22,23]. It reversibly and competitively binds to circulating and clot-bound factor Xa through S1 and S4 pockets, resulting in high selectivity to factor Xa compared to other factors [22,23]. The direct and selective action of rivaroxaban is shown to increase effectivity compared to other indirect factor Xa inhibitors such as VKA or LMWH [5,22,23]. Rivaroxaban also has a high oral bioavailability of 80%, reaches peak plasma concentrations in 2.5 to 4 h, and has no food or drug interactions that necessitate frequent monitoring [3]. Due to its effectivity, 5–40 mg rivaroxaban taken either once daily or in multiple doses for 12–35 days are used in phase-3 clinical trials for thromboprophylaxis for post orthopedic surgery patients [5,22,23,24]. Compared to enoxaparin, previous RCTs have shown reductions in DVT, PE, and mortality in patients consuming rivaroxaban, with rates of 1.1 to 3.7% obtained for post-hip arthroplasty and 9.6% to 18.9% for post-knee arthroplasty [25].

However, rivaroxaban is commonly associated with bleeding side effects, with the same or higher risks of bleeding observed compared to other thromboprophylaxis agents [5,24,25]. One study analyzing prospective registries reported bleeding side effects in 42.9% patients; these were dominantly nonmajor bleeding (58.9%) events, followed by nonmajor clinically relevant bleeding (35.0%) and major bleeding events (6.1%) [25]. However, mortality rates due to bleeding complications were reported in 0.3% of all bleeding events and in 10% of major bleeding events, within which 60% of the patients managed to be treated [25]. Compared to enoxaparin, rivaroxaban has been shown to increase the risk of major bleeding, including GI bleeding, intracranial bleeding, retinal bleeding, epidural hematoma, and adrenal bleeding [22,25,26]. However, the results of bleeding event comparisons between rivaroxaban and other anticoagulant agents still vary due to the different definitions of bleeding used in trials compared to daily practice [5,24,25].

Regarding the results of this meta-analysis, two outcomes were tested: efficacy as measured by the recurrence of VTE, and safety as measured by clinically relevant bleeding. From the 8539 randomized patients included from the five included studies, it was found that the use of rivaroxaban as a thromboprophylaxis agent decreased the occurrence of VTE and all-cause mortality compared to enoxaparin, with a relative risk of 0.38 obtained (95% CI = 0.27–0.54). These results are synergistic with the systematic review study conducted by Nieto et al. [10] that compared the efficacy and safety of Direct Oral Anticoagulant (DOACs) in general with enoxaparin while also including rivaroxaban as one of the drugs tested compared to enoxaparin. There were three included studies, RECORD-1, RECORD-3, and RECORD-4, with a total of 6627 patients included and randomized for efficacy analysis in the form of the recurrence of VTE. The study obtained an RR of 0.19 (95% CI 0.05–0.81), with I^2^ = 73%. Our above analysis found that not only did rivaroxaban reduce the incidence of any VTE and all-cause mortality, but it also significantly reduced the risk of major VTE and proximal and distal DVT [5,7,8].

The additional new insight resulting from this review was the provision of information on the incidence of PE. The administration of rivaroxaban did not provide a significant difference in the incidence of PE compared with enoxaparin. However, new RCT studies are still needed in this regard because the data are quite heterogeneous. In addition, based on the results of the pooled analysis, it was found that rivaroxaban can reduce the incidence of distal (RR 0.53, 95% CI = 0.41–0.68) and proximal DVT (RR 0.2, 95% CI = 0.05–0.73). These findings can provide recommendations for considerations whether rivaroxaban administration has a better effect in preventing DVT than PE. Further, this meta-analysis, after the addition of several recent studies, such as studies by Samama et al., Tang et al., and Xie et al., provided synergistic results with a decrease in the confidence interval range, thereby increasing the precision of the data [11,12,13]. In addition, after adding several additional studies, it was found that the distribution of data became more homogeneous in the two primary outcomes for the efficacy (incidence of VTE and all-cause death) and safety (clinically relevant bleeding and major bleeding) variables, although some variables still had high heterogeneity.

This study included only RCT studies, but other cohort studies also support the results of this meta-analysis. A study by Loganathan et al. showed that a cohort of 479 postoperative hip and knee arthroplasty patients treated with rivaroxaban, with outcomes of PE, DVT, death, stroke, and myocardial infarction (MI) [24]. From the results of this study, VTE, stroke, or MI did not occur, whereas from a safety outcome, there was one (0.2%) patient with major bleeding and nine (1.9%) with non-major bleeding, showing rivaroxaban as an effective anticoagulant for thromboprophylaxis after hip and knee arthroplasty [24]. As explained above, rivaroxaban is not the only DOAC; there are various drugs that are also often used with equally good efficacy. Another cohort study by Alan et al. compared the efficacy of three DOAC drugs, with superior results for rivaroxaban versus dabigatran with apixaban obtained for the prevention of VTE, with *p* < 0.01 for comparison of both. Thus, they found rivaroxaban to be superior to the other two drugs for the prevention of VTE [27].

The results of the safety analysis in this meta-analysis showed that out of a total of 10,879 patients, there were no significant differences in any clinically relevant bleeding events between the group treated with either rivaroxaban or enoxaparin, with a risk ratio of 1.07 (95% CI = 0.9–1.27) obtained. Similar to the major bleeding variable, there was no significant difference between the rivaroxaban and enoxaparin groups, with an RR of 0.97 (95% CI = 0.56–1.68) obtained. The previous systematic review and meta-analysis by Nieto et al. [10] included three RCT studies, RECORD-1, RECORD-3, and RECORD-4, with a total of 9926 patients included, and obtained an RR 1.29 (95% CI =1.03 to 1.63). In this study, the results showed that the administration of rivaroxaban was associated with a higher risk of bleeding than enoxaparin. With the addition of a new study, the results showed that there was no significant difference in clinically relevant bleeding in rivaroxaban compared to enoxaparin. Other cohort studies that support these findings, such as Loganathan et al., found that there was no significant difference between the rivaroxaban group and LMWHs for the parameters of nonmajor bleeding and major bleeding. [24] Another cohort study by Alan et al. showed that rivaroxaban had the same bleeding risk as apixaban and dabigatran, which was around 1.3%; i.e., there were no significant differences between them [27]. Another RCT study by Alok et al. compared the use of rivaroxaban and a placebo in high-risk ambulatory cancer patients [28]. Out of 1080 randomized patients, bleeding occurred in 8 out of 405 patients given rivaroxaban, whereas in the placebo, bleeding occurred in 4 out of 404; the hazard ratio was 1.96 (95% CI = 0.59 to 6.49) [28]. This shows that in terms of safety, rivaroxaban at least has the same risk as LMWHs. Based on the NICE guidelines, rivaroxaban administration has a greater risk of bleeding than enoxaparin, so clinicians need to consider the benefits and ratios obtained when choosing between administering rivaroxaban and enoxaparin. The results of this meta-analysis show that rivaroxaban and enoxaparin do not significantly differ in the occurrence of clinically relevant bleeding or major bleeding. Therefore, it is hoped that this review can provide additional insights to clinicians and guidelines in their considerations. Even though the bleeding risk of rivaroxaban is the same as that of enoxaparin, the efficacy of rivaroxaban is much better compared to enoxaparin.

## 6. Recommendation

Based on the systematic review and meta-analysis that was conducted here, we present the following recommendations:Given that rivaroxaban has better efficacy than enoxaparin, the use of rivaroxaban as a thromboprophylactic agent for the prevention of VTE, especially for the prevention of proximal or distal DVT in adult patients, may be considered.The patient’s bleeding profile should be carefully evaluated before administration of anticoagulant drugs to prevent bleeding incidents.

## 7. Conclusions

This study found that the efficacy of rivaroxaban was superior to enoxaparin. It was statistically demonstrated that rivaroxaban can reduce the incidence of VTE, DVT, and major VTE. In terms of safety parameters, there was no significant difference between rivaroxaban and enoxaparin in the two analyzed variables (major bleeding and clinically relevant bleeding). From this, it can be concluded that, although rivaroxaban has a quite good efficacy, more RCT studies are still needed to provide an assessment of the safety for its use in orthopedic surgery patients.

## Figures and Tables

**Figure 1 jcm-11-04070-f001:**
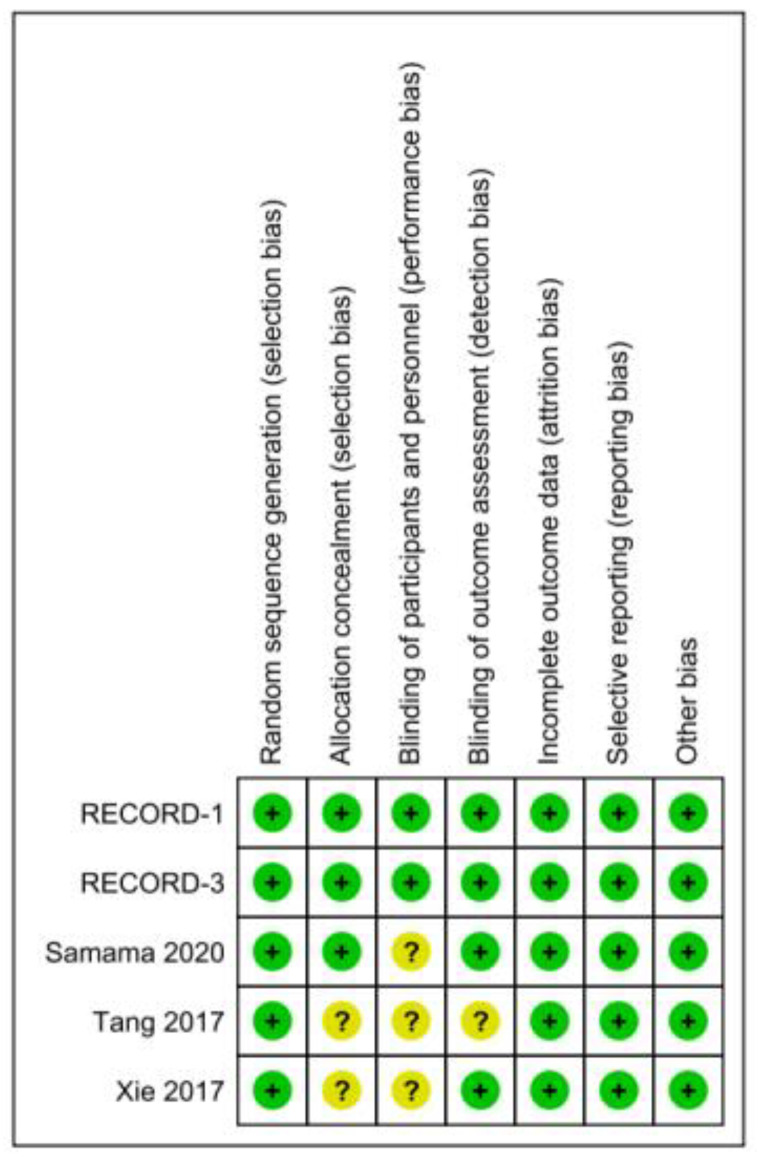
Risk of bias assessment results.

**Figure 2 jcm-11-04070-f002:**
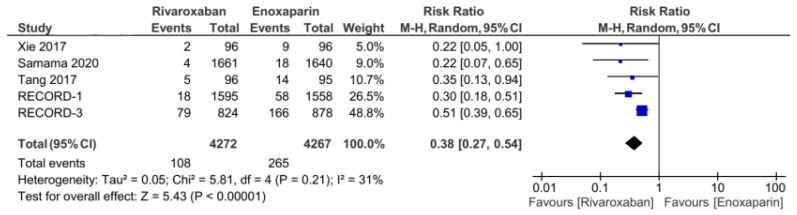
Incidence of any VTE and all-cause death [5,7,12,13,14].

**Figure 3 jcm-11-04070-f003:**
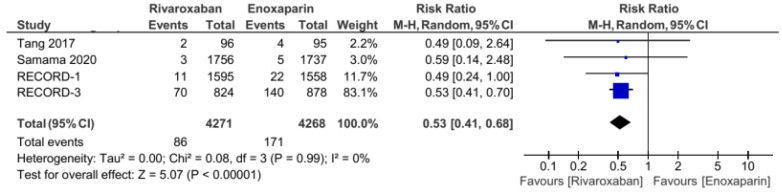
Incidence of distal DVT [5,7,13,14].

**Figure 4 jcm-11-04070-f004:**
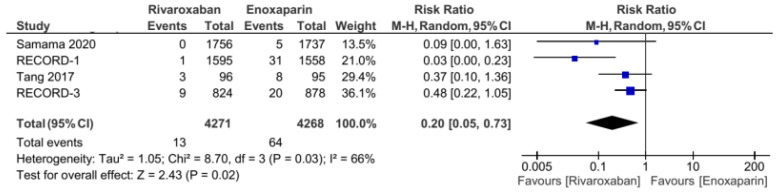
Incidence of proximal DVT [5,7,13,14].

**Figure 5 jcm-11-04070-f005:**
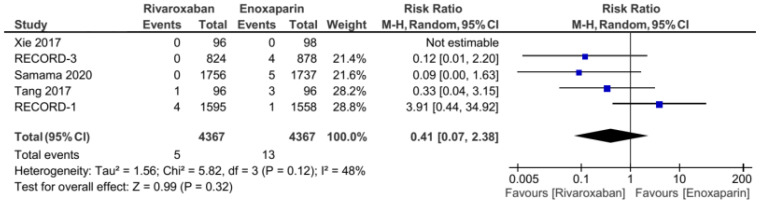
Incidence of fatal and nonfatal PE [5,7,12,13,14].

**Figure 6 jcm-11-04070-f006:**
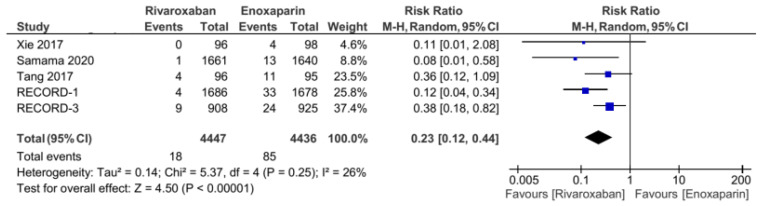
Major VTE [5,7,12,13,14].

**Figure 7 jcm-11-04070-f007:**
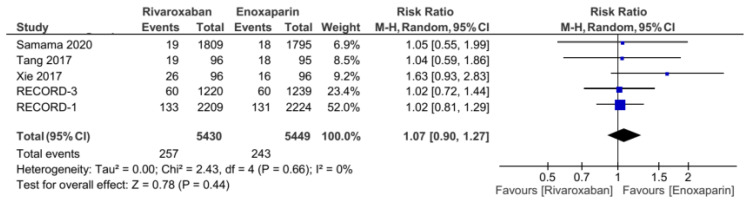
Incidence of any clinically relevant bleeding (major bleeding and any other clinically relevant bleeding) [5,7,12,13,14].

**Figure 8 jcm-11-04070-f008:**
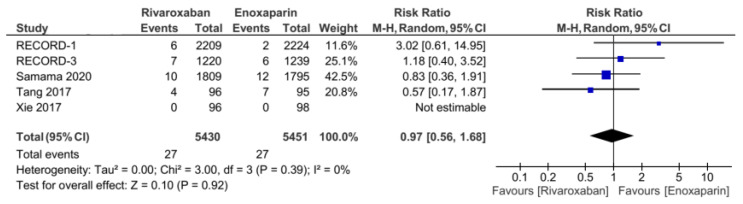
Incidence of major bleeding [5,7,12,13,14].

**Table 1 jcm-11-04070-t001:** Keywords for database searches.

Database	Keyword
PubMed	(((rivaroxaban) AND ((((orthopedic surgery) OR (total knee replacement)) OR (Hip replacement)) OR (Knee replacement))) AND (Enoxaparin)) AND ((((Bleeding) OR (Safety)) OR (Efficacy)) OR (incidence))
SCOPUS	TITLE-ABS-KEY (((rivaroxaban) OR (bay 59–7939)) AND ((orthopedic AND surgery) OR (hip AND replacement) OR (hip AND arthroplasty) OR (knee AND replacement) OR (hip AND arthroplasty)) AND ((enoxaparin) OR (heparin) OR (lmwh)) AND ((bleeding) OR (safety) OR (efficacy) OR (vte AND incidence))) AND (LIMIT-TO (SRCTYPE, “j”)) AND (LIMIT-TO (DOCTYPE, “ar”)) AND (EXCLUDE (PUBYEAR, 2011) OR EXCLUDE (PUBYEAR, 2010) OR EXCLUDE (PUBYEAR, 2009) OR EXCLUDE (PUBYEAR, 2008) OR EXCLUDE (PUBYEAR, 2007) OR EXCLUDE (PUBYEAR, 2006) OR EXCLUDE (PUBYEAR, 2005) OR EXCLUDE (PUBYEAR, 2004)) AND (LIMIT-TO (LANGUAGE, “English”)) AND (EXCLUDE (EXACTKEYWORD, “Apixaban”) OR EXCLUDE (EXACTKEYWORD, “Retrospective Study”) OR EXCLUDE (EXACTKEYWORD, “Comparative Study”) OR EXCLUDE (EXACTKEYWORD, “Retrospective Studies”) OR EXCLUDE (EXACTKEYWORD, “Meta Analysis”) OR EXCLUDE (EXACTKEYWORD, “Systematic Review”) OR EXCLUDE (EXACTKEYWORD, “Case Report”))
EBSCOhost	rivaroxaban AND enoxaparin AND (orthopedic surgery or orthopedic procedure or hip or knee) AND (efficacy or effectiveness or impact or benefits or outcomes or bleeding or incidence)
EMBASE	(‘rivaroxaban’/exp OR rivaroxaban) AND (‘enoxaparin’/exp OR enoxaparin) AND (‘orthopedic surgery’/exp OR ‘orthopedic surgery’ OR ‘knee replacement’/exp OR ‘knee replacement’ OR ((‘knee’/exp OR knee) AND (‘replacement’/exp OR replacement)) OR ‘hip replacement’/exp OR ‘hip replacement’ OR ((‘hip’/exp OR hip) AND (‘replacement’/exp OR replacement))) AND (‘incidence’/exp OR incidence OR ‘bleeding’/exp OR bleeding OR ‘efficacy’/exp OR efficacy OR ‘safety’/exp OR safety) AND (‘clinical trial’/de OR ‘controlled clinical trial’/de OR ‘phase 3 clinical trial’/de OR ‘randomized controlled trial’/de) AND [adult]/lim

**Table 2 jcm-11-04070-t002:** Summary of included studies.

Study	Mean Age Rivaroxaban Group	Mean Age Enoxaparin Group	Country	Population Included	Randomized	Rivaroxaban	Enoxaparin	Duration of Treatment
Xie et al. [12]	65.2 ± 5.5	66.8 ± 7.4	West China	Adults aged 18 years and older, scheduled for primary unilateral total knee replacement for osteoarthritis	196	10 mg orally	40 mg subcutaneous injection	15 days
Samama et al. [13]	41 (29–54)	41 (29–54)	Worldwide	Adults who had undergone nonmajor orthopedic surgery in lower limbs and received thromboprophylaxis (at least 2 weeks)	3604	10 mg orally and injection of placebo	40 mg subcutaneous injection	First 24 h to up to 2 months
Tang et al. [14]	72 ± 14	68 ± 17	China	Admitted to the hospital within 24 h following injury, diagnosed by X-ray and/or CT, and all patients who received internal fixation.	287	10 mg/d at 6 h following operation	40 mg subcutaneous injection, 12 h following the operation	6 h following operation to 28 days
RECORD-1 [5]	63.1 (18–91)	63.3 (18–93)	Worldwide	>18 years who had undergone elective total hip arthroplasty	4541	10 mg orally	40 mg subcutaneous injection	Start at 6 h post-surgery, continued until 35 days
RECORD-3 [7]	67.6 (28–91)	67.6 (30–90)	Worldwide	>18 years of age, knee arthroplasty	2531	10 mg orally	40 mg subcutaneous injection	Start at 6 h post-surgery, continued until 35 days

## Data Availability

The data presented in this study are available on request from the corresponding author.

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
