# Peer review of "Comparison of the Efficacy and Safety of Rivaroxaban and Enoxaparin as Thromboprophylaxis Agents for Orthopedic Surgery—Systematic Review and Meta-Analysis"

_jcm, 2022, doi:10.3390/jcm11144070_

Round 1

Reviewer 1 Report

The manuscript is written on the adequate scientific level. It is prepared on thirteen pages, two of which provide references. The manuscript contains nine figures and two tables.

Content suggestions:

  1. The manuscript does not contain any resulting recommendation for practice. Thus, from the practical point of view, I would like to kindly ask the authors to prepare some suggestions. what can we do on the basis of these findings in the practice...? I would like to kindly ask the authors just to propose how to make their findings helpful for the clinicians,

Author Response

Response to Comments from Reviewer 1

REVIEWER 1 EVALUATION

The manuscript is written on the adequate scientific level. It is prepared on thirteen pages, two of which provide references. The manuscript contains nine figures and two tables.

Content suggestions:

  1. The manuscript does not contain any resulting recommendation for practice. Thus, from the practical point of view, I would like to kindly ask the authors to prepare some suggestions. what can we do on the basis of these findings in the practice...? I would like to kindly ask the authors just to propose how to make their findings helpful for the clinicians,

Thank you very much for the suggestions. We agree with the reviewer that it is important to add recommendations for clinical practice especially for physicians that treat postoperative orthopedic patients. We have added some clinical recommendations regarding the results obtained from this meta-analysis as follows:

“Based on the systematic review and meta-analysis that was conducted here, we present the following recommendations:

  1. We recommend using rivaroxaban as an orthopedic surgical thromboprophylaxis agent for the prevention of VTE, especially in proximal and distal VTE in adult patients, due to its good efficacy.
  2. Since the use of rivaroxaban has the same safety as enoxaparin, we recommend that after administering rivaroxaban, clinicians should monitor the patient's bleeding profile to prevent bleeding incidents.”

Reviewer 2 Report

Dear Editor,

I would like to thank you for giving me the opportunity to review the manuscript titled “Efficacy and safety comparison of Rivaroxaban and Enoxaparin as thromboprophylaxis for orthopedic surgery: a systematic review and meta-analysis”.

The authors, after systematic review of the literature, pooled data from five RCTs revealing that Rivaroxaban was more efficacious than Enoxaparin with regards to safety (mainly VTE occurrence), while it was equally safe.

Introduction:

*This section needs editing with regards to grammar, syntax, and vocabulary, performed by an experienced scientific literature writer.

*The aim (objectives) of the study can be more defined more clearly at the last 1-2 sentences of introduction.

Methods:

*This section also needs editing with regards to grammar, syntax, and vocabulary, performed by an experienced scientific literature writer.

*The algorithms can be presented in the appendix.

*Did the reviewers work independently and blinded the one to each other?

*I would encourage the authors to modify this section by using a state-of-the art systematic review as a template. Please see this as an example.

*Random effect model (not randomised)

Results

*Were the reviewers seven (as stated in the Results) or 6 as stated in Methods? Regardless, this information does not need to be reported in this section.

*232 studies were excluded because they did not comply with PICO as per authors. This is very vague. If these studies or some of these studies were possibly relevant to the studied topic, the exact reasons of exclusion of each study is needed or else there is no other way to demonstrate that there is no selection bias.

Author Response

Response to Comments from Reviewer 2

REVIEWER 2 EVALUATION

  1. *This section needs editing with regards to grammar, syntax, and vocabulary, performed by an experienced scientific literature writer.

Thank You for the comments. We have used the MDPI professional English editing service to improve the English language quality of the manuscript. The aim (objectives) of the study can be more defined more clearly at the last 1-2 sentences of introduction.

Thank you for the input. We have added the aims of this study at the end of the introduction section as suggested by the reviewer. The aim of this systematic review is to provide an update on the efficacy and safety of rivaroxaban for VTE prophylaxis after orthopedic surgery. Due to the existing reviews regarding this issue, it has been quite a while, and the author feels the need for an update because there are several new RCTs. Because more RCTs and, therefore, patients are included in this review than the previous review, more homogeneous data will be provided. We are grateful for the input from the reviewers in improving the manuscript.

*This section also needs editing with regards to grammar, syntax, and vocabulary, performed by an experienced scientific literature writer.

  1. Thank You for the comments. We have used the MDPI professional English editing service to improve the English language quality of the manuscript. *The algorithms can be presented in the appendix.

Thankyou for the comments. We agree completely with the reviewer, and we moved the algorithms to the appendix section as suggested by the reviewer.

  1. *Did the reviewers work independently and blinded the one to each other?

Yes, reviewers work independently and blinded the one to each other. We've revised this issue in the methods section to improve the clarity.

  1. *I would encourage the authors to modify this section by using a state-of-the art systematic review as a template. Please see this as an example.

Thank you for the advice, we have revised the method section format according to the article recommended by the reviewer to us through the editor (https://doi.org/10.7326/M20-8148). From there we've modified the format of the method, so it's currently structured as follows. We sincerely hope the revision is satisfactory and we are open to additional revisions should the reviewer suggests it.

  • Protocol and registration
  • Ethical Approval
  • Literature Search and Data Sources
  • Study Selection
  • Data Abstraction and Risk-of-Bias Assessment
  • Data Synthesis and Analysis
  1. *Random effect model (not randomised)

Thankyou very much for pointing out this issue. We have revised the manuscript based on the suggestions given.

  1. *Were the reviewers seven (as stated in the Results) or 6 as stated in Methods? Regardless, this information does not need to be reported in this section.

Thankyou for the comments, we agree with the reviewers regarding this, so we have removed this section from the results section. In regards to the first question, in this systematic review there are 7 reviewers.

  1. *232 studies were excluded because they did not comply with PICO as per authors. This is very vague. If these studies or some of these studies were possibly relevant to the studied topic, the exact reasons of exclusion of each study is needed or else there is no other way to demonstrate that there is no selection bias.

Thanks for the input, we have added the specific reasons for each study being excluded due to its incompatibility with PICO. Here's what we wrote in the manuscript:

“The initial search for articles in four databases using specific keywords for each database (Table 1) returned 243 articles, 52 of which were duplicates and removed. Abstracts of the remaining 191 studies were subsequently screened. From the abstract screening, 183 studies were excluded because they did not comply with PICO. Specifically, 117 had non-RCT study designs (were cohorts and review articles), the patient criteria were not relevant (nonorthopedic surgery) in 1 study, 10 did not use enoxaparin as the comparator drug used, 37 not using rivaroxaban as an intervention drug, and lastly, in 18 studies, the outcome sought did not match the established PICO (efficacy and safety).”

The changes are also reflected on the revised flowchart

During the revision process, we realized that there was a miscalculation in the total number of reported studies, so we also revised the number of studies obtained. We will really appreciate all the feedback that will be given.

Reviewer 3 Report

This is a very interesting meta-analysis devoted to investigation of the  efficacy and safety comparison of rivaroxaban and enoxaparin as thromboprophylaxis for orthopedic surgery. The article is well written and the conclusions are solid. The authors concluded that the efficacy of rivaroxaban was superior to enoxaparin. 

Author Response

Response to Comments from Reviewer 3

REVIEWER 3 EVALUATION

  1. This is a very interesting meta-analysis devoted to investigation of the efficacy and safety comparison of rivaroxaban and enoxaparin as thromboprophylaxis for orthopedic surgery. The article is well written, and the conclusions are solid. The authors concluded that the efficacy of rivaroxaban was superior to enoxaparin.

Thankyou very much for your praise to our manuscript. We cannot describe how ecstatic we are with words when we read this.

Round 2

Reviewer 2 Report

The authors have addressed my prior comments. 

Author Response

thank you for the revision

thank you for providing input so that the manuscript becomes better